# Social determinants of pain, distress, and quality of life in injured workers: A cross-sectional and longitudinal analysis of patient-reported outcomes

Mohammad Bayattork[1,2], Saghar Soltan Abadi[1], Tom Carter[3], Lynn Cooper[4], David M. Walton[1]*

1 School of Physical Therapy, Western University, Ontario, London, Canada, 2 Department of Sport Sciences and Physical Education, Faculty of Humanities Science, University of Hormozgan, BandarAbbas, Iran, 3 CBI Health Group Inc., Ontario, Canada, 4 Canadian Injured Workers Alliance, Thunder Bay, Ontario, Canada

* dwalton5@uwo.ca

## Abstract

### Background and Objectives

Prognostic tools are widely used to guide early management of musculoskeletal (MSK) injuries. However, social identity factors such as age, gender, race, and socioeconomic status may influence symptom experience and reporting, potentially affecting the accuracy of risk classification. The purpose of this study is to identify those social identity variables associated with scores on common prognostic tools or their prognostic accuracy.

### Methods

We analyzed data from a cohort of workers (n = 203) with acute musculoskeletal injuries. Participants completed the Numeric Pain Rating Scale (NPRS), Traumatic Injuries Distress Scale (TIDS), EuroQoL (EQ5D-5L), and a demographic survey including age, sex, race, income, education, and perceived discrimination. Differences in baseline scores were assessed between identity groups. Recovery was dichotomized (fully recovered/not fully recovered) using 3 different recovery indicators. Predictive accuracy for recovery was evaluated using area under the receiver operating characteristic curve, sample-wide and when stratified by social identity variables.

### Results

Older participants reported significantly higher pain (5.5 vs. 4.3/10, p < 0.01) and those indicating more experiences of discrimination rated higher post-trauma distress (11.1 vs. 9.2/24, p < 0.05). 8-week recovery rates were 21.7% to 54.7%. Lower education predicted <full recovery in administrative data only. The TIDS and EQ5D-5L

**Data availability statement:** All relevant data files are available from the Borealis database, doi 10.5683/SP3/ZIMW1T.

**Funding:** The funder (WSIB) was arm's length and has had no impact on the final results or presentation thereof. The funders had no role in study design, data collection and analysis, decision to publish, or preparation of the manuscript.

**Competing interests:** The authors have declared that no competing interests exist. D. Walton is creator of the Traumatic Injuries Distress Scale, which is available for free download and use through a Creative Commons Attribution-NonCommercial-ShareAlike 4.0 International general use license, meaning it can be freely downloaded, used, and modified as long modifications remain available to the public and neither the original nor adapted versions are being sold or used to generate profit without permission of D. Walton. T. Carter is employed by the CBI Health Group Inc. in Canada, with no part of this project influencing his employment or reimbursement. L. Cooper is employed by the Canadian Injured Workers Alliance, and is a patient partner on the team that, amongst other roles, ensured that the design and dissemination of the project reduced as much as possible the potential for unintended harms to injured workers. S. SoltanAbadi is a PhD trainee and M. Bayattork is a Postdoctoral Associate, both of whom participated in data collection and/or analysis.

were significantly better at predicting recovery compared to the NPRS. When disaggregated by social identity, the TIDS functioned significantly better in females than males, while the EQ5D-5L functioned similarly across all social categories.

### Conclusion

The results indicate that different risk/prognosis cut-scores, and even different tools, may be required for people with different intersectional identities. The results should be interpreted in light of some identities being reduced to broad categories. Most risk/prognosis tools in MSK recovery research use a single universal cut-score to distinguish low from high risk, but this study only partially supports that approach.

---

## Introduction

Work-related physical traumas are a significant public health concern, accounting for about one in every six traumatic injuries in North America [1]. Workers with musculoskeletal (MSK) injuries often endure persistent pain and functional impairment, adversely affecting their quality of life and imposing a significant burden on both workers and employers through lost productivity, high medical costs, and considerable economic losses [2,3]. The complex interaction of physical symptoms, psychological and social factors that influence the severity and persistence of these pain and disability introduce challenge to effective rehabilitation [4–6].

Grounded in the notion that the best approach to managing chronic pain is to prevent it in the first place, one approach to mitigating the impact of chronic post-trauma pain is to optimize acute management [7]. Prognostic tools including questionnaires [8,9] or clinical prediction rules [10,11] are intended to identify the 'high risk' patient for targeted early intervention. Several such tools have been published, including the STarT Back [12], Orebro Musculoskeletal Pain Screening Questionnaire (OMPSQ, [13]) and Traumatic Injuries Distress Scale (TIDS, [14]) all previously demonstrating prognostic accuracy for predicting chronic pain or disability in different samples. Other, often shorter tools have also shown varying levels of prognostic validity, such as the Numeric Pain Rating Scale (NPRS, [15]), self-rated general health [16], or measures of depressive symptom severity [17].

The use of risk stratification tools has been facilitated by the publication of threshold or 'cut' scores, being the scale score that best discriminates between high and low risk groups. As an example, a prior meta-analysis exploring the prognostic validity of the NPRS in acute whiplash associated disorder (WAD) found that a score of 6/10 or greater provides the best discriminative accuracy between high and low risk groups [18]. We have previously reported that a TIDS score <3/24 is optimal for classifying people with acute MSK injuries into a low-risk category, while a score >11/24 is most accurate for identifying the high-risk category [19]. Importantly, to our knowledge few, if any, of these have been explored for differential function between people of different intersectional identities. Using the NPRS as an example, an abundance of evidence indicates that women [20], people of colour [21], or people of lower socioeconomic

status [22] tend to rate higher mean pain intensity for a given stimulus or clinical condition when compared to men, white, and/or wealthier cohorts. If this is consistent, then the rote application of a single risk score to all people, regardless of personal identities or social pressures, may lead to inaccurate estimates of risk in people with acute pain or injury.

As complex computer-assisted models for prognosis and risk-targeted personalized treatment algorithms become more common, it is important to consider what personal or social identity variables should be included in such models. The purpose of this study is to examine differences in baseline values, 8-week prospective outcomes, and prognostic accuracy of three outcome variables, specifically when analyzed across social or demographic subgroups.

## Materials and methods

This hypothesis-generating, cross-sectional and short prospective (8-week follow-up) study is part of a larger project (clinicaltrials.gov ID 06016647) conducted in collaboration with a network of community-based physiotherapy clinics providing Workplace Safety and Insurance Board (WSIB)-funded protocol-driven care in Ontario, Canada. It is reported according to the STROBE (Strengthening the Reporting of Observational Studies in Epidemiology) statement [23]. It was conducted in accordance with principles of ethical research and received approval from the Western University Research Ethics Board (REB) prior to initiating recruitment. All participants provided written electronic (digital) consent prior to providing any data.

### Participants

The recruitment period for this study began on October 18, 2023, and concluded on May 30, 2025. Eligible participants for this study were injured workers aged 18 or older, capable of reading at a 6th-grade English level and receiving physiotherapy for MSK workplace injuries through a network of clinics providing WSIB-funded MSK Program of Care (PoC) treatment. The PoC provides some standardization to the treatment protocols, expected to include any or all of education, pain and self-management strategies, activity modification, exercise therapy, manual therapy, and therapeutic modalities over an 8-week timeline. Workers are eligible for PoC-based care if they are within 8 weeks of an injury affecting soft tissues (e.g., muscle, tendon, ligament) or closed non-displaced fracture not requiring surgical fixation, though some case-by-case flexibility in these criteria is allowed. For our study other exclusion criteria were: self-reported cognitive or reading/comprehension difficulties that could hinder following multi-step instructions, active cancer, infection, neuromuscular diseases, or other major organ disease or comorbidities that would require alternate levels of rehabilitation care.

The recruitment frames for this study were CBI Health Group Inc. clinics in Ontario Canada that provide the WSIB MSK PoC. Recruitment occurred through two main channels: (1) email invitations sent after participants' initial clinic visits, containing study details and a link to the consent form on the Qualtrics® (Seattle, WA) platform, and (2) recruitment posters in clinic waiting areas. In both cases it was up to the worker to determine if they wanted to explore the link and read the Letter of Information. Their participation was not disclosed to WSIB or the treating therapist, and their decision to participate had no effect on the intervention provided or rehabilitation funding. All participants completed an online consent and screening process to ensure eligibility followed by completion of the full study survey. All data were collected and stored on secure servers accessible only to researchers.

### Measurement and instruments

Participants completed a package of questionnaires through the online survey platform. We accessed the survey data for research purposes on June 10, 2025, after the recruitment period concluded. All data were anonymized to ensure participant confidentiality both during and after the study. For the purposes of this analysis, we extracted the following from the main dataset:

#### Predictors

The baseline (intake) dependent variables of interest were pain intensity using the NPRS, quality of life through the EuroQoL 5-Dimension 5-Level version (EQ5D-5L, [24]), and trauma-related emotional distress with the Traumatic Injuries

Distress Scale (TIDS) [14]. The NPRS is a single 0–10 scale assessing pain intensity on average over the past 24 hours with anchors of 0 = no pain and 10 = extreme pain. The EQ5D-5L is a widely used tool for assessing quality of life across five dimensions: mobility, self-care, usual activities, pain/discomfort, and anxiety/depression. We used the more sensitive 5-level version [24] with response options of: no problems, slight problems, moderate problems, severe problems, and extreme problems. Responses were transformed into an EQ5D index, which provides utility scores ranging from 0 (worst QoL) to 1 (best QoL) [25]. The TIDS is a self-reported measure evaluating acute traumatic distress following non-catastrophic injuries. It consists of 12 questions rated on a scale of 0 (never), 1 (sometimes) and 2 (often), yielding a score range from 0 (no distress) to 24 (severe distress). The TIDS has shown strong evidence of concurrent and factorial validity, and significant (>75%) prognostic accuracy in predicting pain and disability up to 12 months later [14,19,26].

## Outcomes

Follow-up data were collected at 4- and 8-week post-entry. Only data at intake and the final (8-week) collection period were used in this study. Three indicators of recovery status were used in this study. Two came directly through participant self-report: 1. Self-rated recovery status ("Overall would you say you have: Fully recovered, Mostly recovered but not fully, Not even close to recovered") dichotomized into Fully Recovered or Not Fully Recovered for analytic purposes; and 2. Current work status dichotomized as: Fully returned to pre-injury work hours and duties, or Not fully returned. The third indicator was drawn from the CBI Health Group Inc. administrative database that tracks, amongst other things, patient outcome at the end of the 8-week PoC in terms of work status or need for additional rehabilitation care. For our purposes, data from the CBI database were coded as 'fully recovered', which included all those fully returned to work and with no further rehabilitation deemed necessary by the treating clinician, or anything less than that – including partial return to work/partial duties, ongoing or alternate level of care required, or PoC extension request. Information on any new injuries experienced since the prior survey was also collected directly from the participants, and those who reported a new injury after entering the study were removed from the prognostic accuracy analysis.

## Social determinants

The social identity variables collected were sex at birth, felt gender, expressed gender, age in years, ethnoracial identity (collected according to the recommendations of the Canadian Institute for Health Information [27]), highest educational level, and annual household income. Table 1 presents the questions asked and response options provided for each variable. An additional variable, the Intersectional Discrimination Index – Major (InDI-M) [28] was collected as an omnibus indication of experiences of unfair treatment or violence as a "result of who you are" (e.g., sex, age, skin colour). It is a 13-item self-report questionnaire that includes experiences such as 'has a healthcare provider ever refused you care?' or 'have you been threatened with a physical or sexual attack?'. Responses are 0 (never), 1 (once) or 2 (more than once) yielding a total score from 0 (no experiences of discrimination) to 26 (many repeated experiences of discrimination). The original development paper [28] supported known-groups validity, sufficient test-retest reliability and evidence of construct validity. We have previously identified a two-factor structure for the InDI-M reflecting two distinct dimensions of discrimination: systemic inequity, and interpersonal violence [29].

## Statistical analyses

As a hypothesis-generating study, analyses were not corrected for multiple comparisons.

## Data cleaning and preparation

Data fidelity was established through visual inspection of distributions or frequencies and values that appeared outside of expected norms. As responses were all collected electronically and downloaded directly there was no risk of data entry

**Table 1. Personal identity and socioeconomic variables collected for this study, and response options presented for each.**

| Variable | Initial Response Options | Categories for analysis |
|---|---|---|
| Sex at birth | Male; Female; Intersex or Undetermined; Prefer not to disclose | Male; Female |
| Felt gender | Mostly or exclusively a man; Mostly or exclusively a woman; Neither exclusively a man or a woman; Indigenous or other cultural gender minority; A gender identity that does not fit any of these options; Prefer not to disclose | N/A[1] |
| Expressed gender | Mostly or exclusively a man; Mostly or exclusively a woman; Neither exclusively a man or a woman; Indigenous or other cultural gender minority; A gender identity that does not fit any of these options; Prefer not to disclose | N/A[1] |
| Ethnoracial Background | Black; East Asian; Indigenous; Latin American; Middle Eastern; South Asian; Pacific Islander; White; Race category not listed (specify); Do not know; Prefer not to disclose | White-identifying; Non-White-identifying |
| Are you perceived or treated as a person of colour in the community where you live or work? | Yes; No; Unsure or prefer not to disclose | Yes; No |
| Annual household income (Canadian dollars) | $0-$20,000; $21,000-$40,000; $41,000-$60,000; $61,000-$80,000; $81,000-$100,000; $101,000-$150,000; $151,000-$200,000; >$200,000 | $0-$60,000; $61,000-$100,000; >$100,000 |
| Highest level of education | Did not finish high school; High school; Vocational or technical school; Community college; Undergraduate university degree; Graduate university degree; Something else (specify) | High school or less; Community college or Vocational school; University undergraduate degree or higher |

1: In only one case was felt or expressed gender different from sex at birth, accordingly these 3 variables were perfectly colinear. To avoid over-fitting the models, only sex at birth was carried through the remaining analyses.

errors by the research team. Normality was evaluated through the Kolmogorov-Smirnov (K-S) test for continuous variables, with significant values ($p < 0.05$) triggering a closer inspection of the data and consideration of how best to manage, including removal of outliers or square-root transformation as the two preferred options. Missing responses were addressed in a two-fold manner for the multinomial questionnaires (TIDS, EQ5D-5L, InDI-M). After ensuring a random pattern of missing data through Little's MCAR test, any questionnaire with ≤20% missing data per respondent had the missing responses imputed through the Expectation-Maximization (EM) algorithm [30]. Where a survey questionnaire had >20% missing responses, or where a response to any of the identity variables was missing, that item was removed from that analysis for that participant. After any necessary transformations or imputations, distributions of responses were examined and it was determined that collapsing some sub-categories into larger response categories was necessary to avoid drawing spurious results (e.g., too few responses in one category). For example, no participant selected 'intersex', so the Sex variable was categorized as male (1) or female (2). Only a single participant endorsed a felt or expressed gender that was different from their sex at birth, meaning those variables were nearly perfectly colinear. Accordingly, only sex at birth was brought forward for further analysis with the interpretation being that the results of that analysis will equally

apply to the gender identity/expression variables. Age, income, and InDI-M were dichotomized into equal-sized groups using a median split, while education was dichotomized into meaningful categories of university undergraduate degree or higher (2) vs. no university degree (1). Owing to the frequency of responses on racial identity, in which no more than 9% identified as non-white, we were forced to dichotomize responses into White-identifying (1) or non-White identifying (2) to retain adequate statistical power. Table 1 presents the final categories.

### Group differences in predictors across sociodemographic variables

Independent t-tests or U-tests were conducted to examine differences in mean NPRS, TIDS, and EQ5D-5L scores across levels of the social determinants. Two-tailed p values were reported as directional hypotheses were not posed at this stage of exploration.

### Group differences in recovery outcomes across sociodemographic variables

All participants who responded to at least one of the follow-up questions (overall recovery, work status) and who did not report a new injury over the 8-week interval between intake and follow-up were dichotomized into fully recovered/not fully recovered or full return to work/less than full return to work. All available administrative data for those who completed the PoC were extracted from the CBI Health Group Inc. database, in which patient status at program completion was coded as either recovered (no further treatment, back to pre-injury work) or not fully recovered (any or all of: modified/partial work duties or hours, further PoC treatment required, alternate level of care required). 2 x 2 tables of proportions were then created for each sociodemographic subgroup (e.g., male/female) and each recovery indicator (e.g., self-rated fully recovered/not fully recovered). As an initial analysis, chi square tests of proportions were conducted to determine whether any sociodemographic subgroup was disproportionately represented within any of the recovery groups.

### Differential effects in discriminating between recovery groups after 8 weeks

To test the ability of each of the 3 potential predictor variables (NPRS, TIDS, EQ5D-5L) to discriminate between those who would later be labelled as recovered or not recovered, receiver operating characteristic (ROC) curves were created for each recovery indicator. Coordinate points on the curve were the ratio of sensitivity to 1-specificity for each available cut-score within the predictor variable. Area under the curve (AUC) was calculated as an omnibus indicator of discrimination accuracy between recovery groups. Differential effects were then tested by creating separate ROCs and calculating a sub-AUC for social identity subgroup (e.g., male/female, younger/older). AUC was compared against chance discrimination as a main effect (lower limit of 95% confidence interval > 0.50) and between the dichotomies of subgroup using standard Z-tests as an indicator of differential discriminative function. Any analysis with fewer than 10 participant in any one cell of the initial 2 x 2 table were excluded from the sub-AUC analysis owing to poor statistical power.

### Sample size

AUC analysis has little firm guidance for sample size and is influenced by the number of predictor and outcome events. We estimated 80% recovery rate based on anecdotal evidence prior to commencing the study, meaning that to achieve a minimum of 10 people in the smallest category, n = 50 per sociodemographic group would be required (combined n = 100) though we targeted higher to account for asymmetrical distributions across some of the variables (e.g., sex).

### Results

Totally, 1,956 injured workers presented to the clinical network for rehabilitation, of which 320 (16.4% of eligible population) people consented to participate. Of those, 203 (63.4% of consenting) provided adequately complete data for this analysis. The final sample was 59% female, mean age 43.9 years (range 20–73), and 61.6% identified as white only. The

 

mean time from injury to completion of the intake survey was 49 days (range 2–90). At 8-week follow-up, 150 provided responses to at least one of the two recovery questions, of which 8 also indicated a new injury in the intervening 8 weeks and were removed. This left n = 142 (70.0% retention) for the AUC analyses using the self-reported recovery data, and we were able to match n = 190 (93.4% retention) with the CBI Health Group Inc. administrative database for the administrative recovery analysis. Comparing completers to non-completers, the only significant differences were for sex and race, with completers more likely to be female (65% vs. 45%) and to identify as white (67.4% vs. 48.4%) compared with non-completers. There were no other significant between-group differences.

### Data fidelity and cleaning

Of the 203 participants there were a total of 2.8% missing responses across all questions at baseline. Across the three predictor variables, 2.4% of questionnaires were missing >20% of responses and needed to be removed, while 0.4% were missing ≤20% of responses that were imputed. Little's MCAR test indicated that after removal the remaining data were missing at random (MCAR p > 0.05). Normality tests using the Kolmogorov–Smirnov test (K-S) indicated that pain scores were normally distributed (K-S = 0.06, p > .05), while TIDS (K-S = 0.07, p < .05) and EQ5D-5L scores were not (K-S = 0.15, p < .05). The EQ5D-5L and TIDS scores were square-root transformed which successfully normalized the distributions for between-group comparisons.

### Differences in pain, quality of life, and distress by social determinants

Older participants (>41 years) reported significantly higher mean pain intensity (mean = 5.5, 95%CI 5.0 to 5.9) compared to younger participants (mean = 4.3, 95%CI 3.9 to 4.7; p < 0.01). Those who reported higher experiences of intersectional discrimination (InDI-M > 3) scored significantly higher on the TIDS (mean = 11.1, 95%CI 10.0 to 12.1) compared to those who reported fewer experiences (mean = 9.2, 95%CI 8.0 to 10.4; p < 0.05). There were no other statistically significant group-based differences at baseline (Table 2).

### Sociodemographic group-based differences in recovery outcomes

Recovery rates differed substantially based on the definition used. When asked directly, only 21.7% (30/138) responded 'yes' to the question about being fully recovered. When respondents were asked about their work status in relation to that held pre-injury, 47.9% (68/142) indicated they had returned to full pre-injury job duties. When querying the CBI Health Group Inc. administrative database, 54.7% (104/190) were considered fully back to work and discharged from further rehabilitation based on clinician or funder opinion. Table 3 presents the proportions of recovery status both overall and stratified by sociodemographic subgroups. Chi square analysis revealed only one difference in proportions that reached Bonferroni-corrected significance: those without a university education were less likely to be coded as fully recovered in the administrative data compared to those with a university undergraduate degree or higher ($\chi^2$ = 7.60, p = 0.006). There were no other differences in proportions recovered across any of the outcomes or sociodemographic categories.

### Risk stratification accuracy

Tables 4–6 present the results of the AUC analyses. For self-rated recovery as the outcome, all three predictor tools showed significant main effect to discriminate between those who did and did not describe themselves as recovered 8 weeks later (NPRS: AUC = 0.66, 95%CI 0.56 to 0.76; TIDS: AUC = 0.75, 95%CI 0.64 to 0.86; EQ-5D-5L: AUC = 0.78, 95%CI 0.70 to 0.87). AUC of both the TIDS and EQ-5D-5L was significantly greater than the NPRS for this outcome. Main effects for discriminating between those who reported being fully returned to pre-injury work status compared to those not fully returned were significant for only the TIDS (AUC = 0.68, 95%CI 0.59 to 0.76) and EQ-5D-5L (AUC = 0.73, 0.65 to 0.82) but not the NPRS (AUC = 0.59, 95%CI 0.49 to 0.68). When administrative data on recovery status at completion

**Table 2. Mean scores and comparisons of difference in each of the three predictor variables by level of sociodemographic identity variable.**

|  | N | NPRS (mean, 95%CI) | TIDS (mean, 95%CI) | EQ-5D-5L (95%CI) |
|---|---|---|---|---|
| Age |  |  |  |  |
| ≤41 years | 95 | **4.3 (3.9, 4.7)** | 9.5 (8.3, 10.6) | 0.66 (0.62, 0.70) |
| >41 years | 100 | **5.5 (5.0, 5.9)**\*\* | 10.7 (9.7, 11.7) | 0.62 (0.58, 0.66) |
| Sex |  |  |  |  |
| Female | 113 | 5.0 (4.6, 5.3) | 10.6 (9.4, 11.7) | 0.63 (0.60, 0.68) |
| Male | 78 | 4.7 (4.3, 5.3) | 9.7 (8.6, 11.0) | 0.63 (0.58, 0.68) |
| Skin Colour (Racial Identity) |  |  |  |  |
| White | 124 | 4.7 (4.4, 5.1) | 9.9 (8.9, 10.8) | 0.64 (0.60, 0.67) |
| Non-White | 74 | 5.3 (4.7, 5.7) | 11.1 (9.6, 12.7) | 0.62 (0.56, 0.67) |
| Education |  |  |  |  |
| No university degree | 101 | 5.0 (4.6, 5.4) | 10.4 (9.3, 11.6) | 0.62 (0.57, 0.66) |
| University degree | 77 | 4.8 (4.3, 5.3) | 9.5 (8.2, 10.6) | 0.65 (0.61, 0.70) |
| Income |  |  |  |  |
| ≤$60,000 | 84 | 5.0 (4.6, 5.4) | 9.9 (8.6, 11.2) | 0.64 (0.59, 0.68) |
| >$60,000 | 98 | 4.9 (4.8, 5.3) | 10.4 (9.3, 11.5) | 0.64 (0.59, 0.68) |
| InDI-M score |  |  |  |  |
| Low (≤3) | 93 | 4.9 (4.4, 5.3) | **9.2 (8.0, 10.4)** | 0.65 (0.61, 0.69) |
| High (>3) | 96 | 4.9 (4.5, 5.3) | **11.1 (10.0, 12.1)**\* | 0.62 (0.58, 0.66) |

**Bolded** are significant differences between the two levels of sociodemographic variable through independent samples t-test. \* = p < 0.05, \*\* = p < 0.01. NPRS = Numeric Pain Rating Scale; TIDS = Traumatic Injuries Distress Scale; EQ-5D-5l = EuroQoL 5-dimension 5-level; InDI-M = Intersectional Discrimination Index – Major.

**Table 3. Proportions of recovery status both overall and when stratified by sociodemographic subgroups.**

|  | Fully recovered (self-reported) | | Fully back to work (self-reported) | | Good treatment outcome (administrative data) | |
|---|---|---|---|---|---|---|
|  | Proportion | $X^2$ (p) | Proportion | $X^2$ (p) | Proportion | $X^2$ (p) |
| Full sample | 21.7% | N/A | 47.9% | N/A | 54.7% | N/A |
| Age |  |  |  |  |  |  |
| ≤41 years (63) | 18 (28.6%) | 3.68 (0.06) | 34 (51.5%) | 0.69 (0.41) | 38 (57.6%) | <0.01 (0.99) |
| >41 years (73) | 11 (15.1%) |  | 32 (44.4%) |  | 42 (57.5%) |  |
| Sex |  |  |  |  |  |  |
| Male (48) | 10 (20.8%) | 0.01 (0.92) | 25 (51.0%) | 0.31 (0.58) | 30 (61.2%) | 0.42 (0.52) |
| Female (88) | 19 (21.6%) |  | 41 (46.1% | | 50 (55.6%) |  |
| Racial identity |  |  |  |  |  |  |
| White (96) | 20 (20.8%) | 0.15 (0.70) | 46 (47.9%) | <0.01 (0.99) | 56 (57.7%) | 0.01 (0.91) |
| Non-white (42) | 10 (23.8%) |  | 22 (47.8%) |  | 27 (58.7%) |  |
| Education |  |  |  |  |  |  |
| No university (72) | 16 (22.2%) | 0.01 (0.94) | 33 (45.8%) | 0.33 (0.57) | **36 (48.6%)** | **7.60 (<0.01)** |
| University degree (57) | 13 (22.8%) |  | 30 (50.8%) |  | **42 (72.4%)** |  |
| Income |  |  |  |  |  |  |
| ≤$60,000 (59) | 16 (27.1%) | 1.76 (0.19) | 29 (48.3%) | 0.04 (0.85) | 34 (56.7%) | 0.04 (0.84) |
| >$60,000 (74) | 13 (17.6%) |  | 35 (46.7%) |  | 45 (58.4%) |  |
| InDI-M score |  |  |  |  |  |  |
| ≤3 (63) | 17 (27.0%) | 2.24 (0.13) | 34 (53.1%) | 1.34 (0.25) | 39 (60.9%) | 0.56 (0.46) |
| >3 (73) | 12 (16.4%) |  | 32 (43.2%) |  | 41 (54.7%) |  |

**Bolded** = difference in proportions is significant between the two groups (via $\chi^2$)

**Table 4. Discriminative accuracy of NPRS, TIDS, and EQ-5D-5L for predicting self-rated recovery status.**

| | Fully Recovered[1] | Not Fully Recovered[1] | NPRS AUC (95%CI) | AUC diff. (95%CI) | TIDS AUC (95%CI) | AUC diff. (95%CI) | EQ-5D-5L AUC (95%CI) | AUC diff. (95%CI) |
|---|---|---|---|---|---|---|---|---|
| Overall | 30 | 108 | **0.66 (0.56, 0.76)** | --- | **0.75 (0.64, 0.86)** | --- | **0.78 (0.70, 0.87)** | --- |
| Age | | | | 0.15 (−0.06, 0.36) | | 0.19 (−0.04, 0.42) | | 0.08 (−0.09, 0.26) |
| ≤41 | 18 | 45 | **0.72 (0.58, 0.86)** | | **0.82 (0.70, 0.94)** | | **0.74 (0.60, 0.87)** | |
| >41 | 11 | 61 | 0.57 (0.42, 0.72) | | 0.63 (0.44, 0.83) | | **0.82 (0.71, 0.93)** | |
| Sex | | | | 0.11 (−0.12, 0.33) | | 0.17 (−0.09, 0.43) | | 0.13 (−0.06, 0.32) |
| Male | 10 | 37 | 0.57 (0.39, 0.76) | | 0.63 (0.40, 0.87) | | **0.70 (0.54, 0.87)** | |
| Female | 19 | 69 | **0.68 (0.56, 0.81)** | | **0.80 (0.69, 0.92)** | | **0.83 (0.74, 0.93)** | |
| Racial Identity | | | | 0.02 (−0.19, 0.23) | | 0.06 (−0.16, 0.27) | | 0.09 (−0.08, 0.25) |
| White | 20 | 75 | **0.67 (0.55, 0.80)** | | **0.73 (0.59, 0.87)** | | **0.75 (0.63, 0.86)** | |
| Non-White | 10 | 33 | 0.65 (0.48, 0.82) | | **0.79 (0.62, 0.96)** | | **0.83 (0.71, 0.95)** | |
| Education | | | | 0.10 (−0.11, 0.32) | | 0.13 (−0.09, 0.36) | | 0.06 (−0.12, 0.23) |
| Less than University | 16 | 55 | **0.70 (0.56, 0.84)** | | **0.81 (0.69, 0.93)** | | **0.82 (0.71, 0.92)** | |
| University degree | 13 | 44 | 0.59 (0.44, 0.75) | | 0.68 (0.48, 0.87) | | **0.76 (0.62, 0.90)** | |
| Income | | | | 0.19 (−0.02, 0.39) | | 0.21 (−0.01, 0.42) | | 0.13 (−0.04, 0.31) |
| ≤$60k | 16 | 42 | **0.75 (0.61, 0.89)** | | **0.85 (0.75, 0.96)** | | **0.87 (0.77, 0.96)** | |
| >$60k | 13 | 61 | 0.56 (0.42, 0.71) | | 0.65 (0.46, 0.84) | | **0.73 (0.58, 0.88)** | |
| InDI-M | | | | 0.02 (−0.19, 0.23) | | 0.14 (−0.07, 0.34) | | 0.04 (−0.14, 0.22) |
| Low (≤3) | 17 | 45 | **0.66 (0.51, 0.81)** | | **0.69 (0.52, 0.86)** | | **0.80 (0.68, 0.91)** | |
| High (>3) | 12 | 61 | 0.64 (0.49, 0.79) | | **0.83 (0.71, 0.94)** | | **0.76 (0.62, 0.89)** | |

1: Fully recovered=A response of 'mostly or fully recovered' to the question "In your opinion, have you fully recovered from your injury?"; Not fully recovered=any response other than 'mostly or fully recovered'. **Bolded**=AUC is significantly greater than chance (lower limit of 95% confidence interval>0.50).

**Table 5. Discriminative accuracy of NPRS, TIDS, and EQ-5D-5L for predicting self-rated work status.**

| | Fully Returned to Work[1] | Not Fully Returned to Work[1] | NPRS AUC (95%CI) | AUC diff. (95%CI) | TIDS AUC (95%CI) | AUC diff. (95%CI) | EQ-5D-5L AUC (95%CI) | AUC diff. (95%CI) |
|---|---|---|---|---|---|---|---|---|
| Overall | 68 | 74 | 0.59 (0.49, 0.68) | | **0.68 (0.59, 0.76)** | | **0.73 (0.65, 0.82)** | |
| Age | | | | 0.18 (−0.01, 0.37) | | 0.04 (−0.14, 0.22) | | 0.09 (−0.08, 0.26) |
| ≤41 | 34 | 32 | **0.69 (0.57, 0.82)** | | **0.66 (0.53, 0.79)** | | **0.68 (0.55, 0.81)** | |
| >41 | 32 | 39 | 0.51 (0.37, 0.65) | | **0.70 (0.58, 0.82)** | | **0.77 (0.66, 0.88)** | |
| Sex | | | | 0.08 (−0.12, 0.29) | | **0.32 (0.13, 0.52)[2]** | | 0.10 (−0.08, 0.29) |
| Male | 25 | 23 | 0.53 (0.36, 0.70) | | 0.47 (0.30, 0.63) | | **0.67 (0.52, 0.83)** | |
| Female | 41 | 48 | 0.61 (0.49, 0.73) | | **0.79 (0.69, 0.88)** | | **0.78 (0.68, 0.88)** | |
| Racial Identity | | | | 0.18 (−0.02, 0.39) | | 0.09 (−0.09, 0.28) | | 0.10 (−0.08, 0.28) |
| White | 46 | 49 | **0.65 (0.54, 0.76)** | | **0.65 (0.54, 0.77)** | | **0.76 (0.67, 0.86)** | |
| Non-White | 22 | 24 | 0.47 (0.30, 0.64) | | **0.75 (0.60, 0.89)** | | **0.67 (0.51, 0.83)** | |
| Education | | | | 0.01 (−0.20, 0.20) | | 0.01 (−0.18, 0.19) | | 0.04 (−0.14, 0.21) |
| Less than University | 33 | 38 | 0.58 (0.45, 0.72) | | **0.68 (0.55, 0.81)** | | **0.75 (0.64, 0.86)** | |
| University degree | 30 | 29 | 0.58 (0.44, 0.73) | | **0.68 (0.55, 0.82)** | | **0.71 (0.58, 0.85)** | |
| Income | | | | 0.12 (−0.08, 0.31) | | 0.02 (−0.16, 0.21) | | 0.07 (−0.10, 0.24) |
| ≤$60k | 29 | 30 | **0.67 (0.53, 0.81)** | | **0.69 (0.55, 0.83)** | | **0.77 (0.65, 0.89)** | |
| >$60k | 35 | 40 | 0.55 (0.42, 0.68) | | **0.67 (0.55, 0.79)** | | **0.70 (0.58, 0.82)** | |
| InDI-M | | | | 0.08 (−0.11, 0.28) | | 0.01 (−0.17, 0.19) | | 0.04 (−0.13, 0.21) |
| Low (≤3) | 34 | 29 | 0.64 (0.50, 0.78) | | **0.67 (0.54, 0.81)** | | **0.75 (0.63, 0.87)** | |
| High (>3) | 32 | 42 | 0.55 (0.42, 0.69) | | **0.68 (0.56, 0.80)** | | **0.71 (0.59, 0.83)** | |

1: Fully returned to work=self-disclosed return to full pre-injury hours and duties; Not fully returned to work=self-disclosed, partial hours or duties or less compared to pre-injury work. **Bolded**=AUC is significantly greater than chance (lower limit of 95% confidence interval>0.50); 2=Difference in AUC is significant between males and females for the TIDS only.

**Table 6. Discriminative accuracy of NPRS, TIDS, and EQ-5D-5L for predicting administrative discharge status[1].**

| | Full Recovery[1] | Not Full Recovery[1] | NPRS AUC (95%CI) | AUC diff. (95%CI) | TIDS AUC (95%CI) | AUC diff. (95%CI) | EQ-5D-5L AUC (95%CI) | AUC diff. (95%CI) |
|---|---|---|---|---|---|---|---|---|
| Overall | 104 | 86 | **0.65 (0.57, 0.73)** | | **0.64 (0.56, 0.72)** | | **0.64 (0.57, 0.72)** | |
| Age | | | | | | | | |
| ≤41 | 59 | 33 | **0.72 (0.61, 0.83)** | 0.17 (0.10, 0.34) | **0.69 (0.58, 0.80)** | 0.13 (−0.04, 0.29) | **0.69 (0.57, 0.80)** | 0.10 (−0.06, 0.27) |
| >41 | 44 | 47 | 0.54 (0.42, 0.67) | | 0.56 (0.44, 0.68) | | 0.59 (0.47, 0.71) | |
| Sex | | | | | | | | |
| Male | 44 | 32 | **0.71 (0.60, 0.83)** | 0.12 (−0.05, 0.28) | **0.65 (0.53, 0.78)** | 0.03 (−0.14, 0.19) | **0.71 (0.59, 0.82)** | 0.11 (−0.05, 0.27) |
| Female | 59 | 48 | 0.60 (0.49, 0.71) | | **0.63 (0.52, 0.73)** | | 0.60 (0.49, 0.71) | |
| Racial Identity | | | | | | | | |
| White | 69 | 49 | **0.67 (0.57, 0.77)** | 0.05 (−0.11, 0.22) | **0.64 (0.54, 0.74)** | 0.01 (−0.16, 0.17) | **0.63 (0.53, 0.73)** | 0.04 (−0.12, 0.20) |
| Non-White | 35 | 37 | 0.62 (0.49, 0.75) | | **0.63 (0.50, 0.77)** | | **0.67 (0.55, 0.80)** | |
| Education | | | | | | | | |
| Less than University | 54 | 41 | **0.65 (0.54, 0.76)** | 0.03 (−0.14, 0.20) | **0.65 (0.54, 0.75)** | 0.01 (−0.16, 0.17) | **0.64 (0.53, 0.75)** | 0.01 (−0.16, 0.17) |
| University degree | 43 | 35 | 0.62 (0.49, 0.75) | | **0.64 (0.51, 0.76)** | | **0.65 (0.52, 0.77)** | |
| Income | | | | | | | | |
| ≤$60k | 47 | 37 | 0.63 (0.51, 0.75) | 0.08 (−0.09, 0.24) | **0.66 (0.54, 0.77)** | 0.02 (−0.14, 0.18) | 0.61 (0.49, 0.73) | 0.048 (−0.08, 0.25) |
| >$60k | 53 | 41 | 0.71 (0.60, 0.81) | | **0.64 (0.52, 0.75)** | | **0.69 (0.59, 0.80)** | |
| InDI-M | | | | | | | | |
| Low (≤3) | 53 | 39 | **0.64 (0.52, 0.76)** | 0.03 (−0.14, 0.19) | **0.64 (0.53, 0.75)** | 0.01 (−0.16, 0.16) | **0.66 (0.55, 0.78)** | 0.03 (−0.13, 0.19) |
| High (>3) | 50 | 42 | **0.67 (0.55, 0.78)** | | **0.64 (0.53, 0.75)** | | **0.63 (0.52, 0.75)** | |

1: Recovery status derived from CBI Health Group administrative data. 'Full recovery' = an amalgam of no further intervention required and/or fully returned to pre-injury work status, typically determined by the treating clinician or insurance provider; 'Not full recovery' = the injured worker either requires an extension of PoC care, transition to an alternate level of care, or work accommodations as of the end of the initial 8-week treatment interval. **Bold** = Discriminative accuracy (Area Under the Curve) is significantly greater than chance (0.50) at the p < 0.05 level.

of the PoC was the outcome, all three tools functioned almost identically and significantly better than chance (NPRS: AUC = 0.65, 95%CI 0.57 to 0.73; TIDS: AUC = 0.64, 95%CI 0.56 to 0.72; EQ-5D-5L: AUC = 0.64, 95%CI 0.57 to 0.72).

When split by subgroups no comparison included any single cell with <10 participants. When the outcome was self-rated recovery, the main effects for both the NPRS and TIDS were only significantly better than chance for younger participants but not older, females but not males, those without a university education but not those with, and those with lower but not higher annual household income (Table 4). The NPRS also showed significant discriminative accuracy only for participants who identified as white but not for non-white, and with a lower InDI-M score but not a higher score. Post-hoc Z-tests indicated none of the AUC differences reached between-group statistical significance despite the significant main effects. The EQ-5D-5L functioned significantly better than chance across all subgroup analyses.

When the outcome was self-rated work status (fully returned / not fully returned), the NPRS was marginally better than chance accuracy for younger participants, those identifying as white, and those with lower household income, while showing no discriminative accuracy in older participants, those identifying as non-white, and those with higher household income (Table 5). Again, post-hocs indicated the magnitude of AUC was not significantly different between any of the subgroups. The TIDS showed significant main effects across all analyses except sex where both main effects and post-hocs indicated AUC was significant in females (AUC = 0.79, 95%CI 0.69 to 0.88) but not males (AUC = 0.47, 95%CI 0.30 to 0.63) with the difference reaching significant (p < 0.01). Again, the EQ5D-5L showed similar and significant AUC across all subgroup analyses.

When administrative discharge status was the outcome (Table 6), the NPRS showed significant discriminative accuracy in only the younger, male, white, less educated, and in both InDI-M groups, but no significant effect in older, female,

non-white, higher-educated, and both income category groups. Post-hoc analysis indicated significantly better function in the younger (AUC = 0.72, 95%CI 0.61 to 0.83) compared to older (AUC = 0.54, 95%CI 0.42 to 0.67) groups (p < 0.01). The TIDS showed consistent and significant discriminative accuracy across all subgroup analyses save for older participants in which it failed to function better than chance accuracy (AUC = 0.56, 95%CI 0.44 to 0.68). The EQ5D-5L demonstrated better than chance accuracy across most subgroup analyses but failed to reach significance in the older (AUC = 0.59, 95%CI 0.47 to 0.71), female (AUC = 0.60, 95%CI 0.49 to 0.71) and lower income (AUC = 0.61, 95%CI 0.49 to 0.73) groups.

## Discussion

The purpose of this study was to contribute to the growing discourse around personalized rehabilitation and prognosis in acute injury by exploring differences in baseline mean values, 8-week prospective outcomes, and discriminative (prognostic) accuracy across 3 relevant predictor and outcome variables when disaggregated by meaningful social identity or demographic subgroups. The results of these analyses provide direction for the development of future prognostic tools in adults (workers) with acute MSK injuries by providing evidence for the sociodemographic variables that can be expected to result in differential prognostic functioning.

Both the NPRS and TIDS have previously shown ability to predict recovery outcomes in acute MSK trauma [18,19] but to our knowledge the finding that the EQ5D-5L index score also functions as a prognostic screening tool in acute MSK trauma is novel. Importantly, in no prior analysis of any of these tools or of other prognostic screening tools has consideration been given to the potential for such tools to function differently, or require different risk cut scores, across different social or demographic strata. While we found little evidence that the outcomes could be simply predicted by sociodemographic variables (university education for predicting administrative outcome being the only exception), or that scores on the three prognostic tools are different (age for the NPRS and experiences of discrimination for the TIDS being the exceptions), we found several examples of potential differential functioning of the prognostic tools between sociodemographic subgroups. Assuming these findings hold in other samples, the most intuitive interpretation is that while there may be little need for different risk cut-scores on prognostic tools, there appears to be a need for deeper consideration about what prognostic tools to use for whom.

It is interesting that of the three tools that we explored, it is the one least aligned with prognosis and with the least prior exploration of prognostic utility (EQ5D-5L) that showed the most consistent accuracy across the three outcomes being predicted, and that its function was largely robust to differences in sociodemographic subgroups. That the NPRS showed the least robust accuracy across outcomes and sociodemographic characteristics is perhaps not surprising given its single-item nature. The TIDS functioned similarly to the EQ-5D-5L though with some notable exceptions. We note that of the three, the TIDS is the only one that was initially developed specifically for the purposes of risk/prognosis screening in acute MSK trauma and can also provide scores on three subscales (uncontrolled pain, negative affect, intrusion/hyperarousal) that are intended to be useful for prioritizing treatment targets in those deemed at higher risk of poor recovery. This added utility may still place it as the most desirable of the three tools explored here, though the function of the EQ5D-5L in this context appears to support further prognostic exploration.

The comparison of mean difference in the three predictors at intake between sociodemographic subgroups reveals interesting results both in terms of the significant differences and the non-significant differences. That the older participants scored on average 1.2 NPRS points higher than the younger participants is supported by considerable prior evidence [31], though does bring into question our own prior meta-analysis findings in which an NPRS score ≥6/10 was best for discriminating between high and low risk of non-recovery after acute neck trauma (whiplash, [18]). Based on existing data at the time, there was insufficient published evidence to suggest the 6/10 threshold should be different for older vs. younger respondents, but the findings here suggest that a single threshold applied equally across all patients may lead to inaccurate prognoses in at least a subset of the population. Despite prior work suggesting females tend to rate higher NPRS than males [32] we did not find that same effect in our data, which is difficult to interpret but may be due to

differences in patient populations (acutely injured workers), sampling frames (community-based insurance-funded physical rehabilitation clinics), or sample size. The non-significant differences across levels of education, income, or ethnoracial identity provide some support for use of a single cut-score on these tools, though we acknowledge that the strata used here are quite broad because of low proportions for many identities. This is particularly problematic in the ethnoracial comparisons, where we were forced to collapse the responses into overly broad 'white' and 'non-white' categories owing to the low frequency of non-white identities in this sample – for purposes of both statistical power and to avoid re-identification of those in smaller categories. While we believe race-based analyses can be valuable when performed with appropriate attention to equity and have included those results here in part as an illustration of the practice, we must acknowledge that the overly-broad categorizations used very likely obscure differential effects for many racial identities. We have provided the full breakdown of ethnoracial identities in Supplemental S1 Table for illustrative purposes. These analyses should not be considered definitive evidence that there is or is not a race-based effect on prognostic tools, rather we have also chosen to present these results in this way in the hopes of opening conversation around equity in research. While calls for sex- and gender-based analyses [33] have contributed to more concerted efforts to recruit women and females into research, there has been less attention paid to recruiting different racialized groups. Perhaps our findings, that did not include special considerations for specifically recruiting non-white participants, will contribute to that conversation.

The only other significant finding was that those workers who indicated a higher number of prior experiences of intersectional discrimination also rated higher mean trauma-related distress on the TIDS. This study was not designed to explore mechanisms so any such commentary is speculative, though assuming experiences of discrimination are stressors this finding would fit with Cumulative Stress Theory (CST, [34]). CST posits a cumulative effect of stressors over a lifetime, such that subsequent stressors (like work injury) are experienced as more distressing by those who have experienced considerable prior stress. Of note is that the mean TIDS score of 11.1 in the high-discrimination group is at the previously published cut-score of 11 for higher risk of chronicity based on data from other cohorts [19]. This appears to align with prior work showing that the strongest predictors of poor outcome after acute injury are not the magnitude of damage or injury but rather the magnitude of distress experienced [35]. The potential for complex interactions or mutual synergy between prior life stressors and new trauma for predicting future recovery introduces an intriguing direction for further research. One such direction is the effect of integrating trauma-informed/assumed care or other strategies that manage cumulative psychological and social load in supporting recovery after injury, beyond focusing solely on the tissues at fault [36].

The data on recovery rates are also of interest and reveal important considerations for how such outcomes are defined. The concept of recovery as simply 'a return to pre-injury state' has long been challenged by scholars and patients alike [e.g., [37]] and discourse on the optimal recovery indicator for quantitative research remains unresolved. Using data from this study, when participants were directly asked whether they believed themselves to be fully recovered, without further delineation of what recovery means to each person, only 21.7% indicated they were fully or mostly recovered. When respondents were directly queried about their work status in relation to pre-injury status, 47.9% indicated they were performing their job duties at the same level. When recovery status was inferred from clinical administrative data, typically being a result of clinician interpretation in partnership with the patient (worker) and insurer, 54.7% were deemed recovered (fully back to work, no additional rehabilitation needed or offered). While recovery rates were similar when work status was explicit within the definition, those rates were considerably lower when recovery was defined by broader personal perspective. One logical interpretation is that people were returning to full work duties before feeling fully recovered – maybe not inherently problematic yet still illustrating the importance of clear and judicious selection of recovery indicators. This appears to have important implications for future prognostic work and will contribute to the ongoing discourse around the nature of recovery and how the concept can best be operationalized.

The results of the differential functioning of the three tools for discriminating between those who would and would not be deemed recovered after an 8-week standardized rehabilitation program require further confirmation though if replicated

these findings create several potentially new lines of inquiry. As we anticipate prognostic tools to become increasingly capable especially in the context of machine learning tools, knowing what variables are important to collect and use in risk stratification is important. To start, both the TIDS and EQ-5D-5L were more accurate in discriminating between recovery groups than the NPRS, and both tools were more accurate when the outcomes were self-reported than when drawn from administrative data. That the EQ-5D-5L functioned largely the same regardless of sociodemographic identity suggests it is worth further exploration in independent cohorts as a potential prognostic tool after MSK injury. The TIDS also functioned largely consistent across sociodemographic strata though there were some subgroups (older age, males, university educated, higher income) for which it failed to discriminate between recovery groups at a level greater than chance for some of the outcomes. While potentially a function of lower-than-expected recovery rates and small numbers in some cells of the 2x2 table introducing the potential for imprecision, this suggests that distress as a predictor of outcome may be most suited for use in younger people, females, those without university degrees and those with lower annual income. If replicated, this would indicate that the type of prognostic tool selected for an individual patient should be at least partly influenced by social and demographic identities of the respondent. However, the only comparison in which the AUC for the TIDS was significantly different between strata was seen when self-reported work status was the outcome and the sample was disaggregated by sex. In that analysis, the TIDS accurately discriminated between recovery groups in females but not males. This finding differs from our prior analysis of TIDS functioning by sex in an independent sample of n = 224 acutely injured adults from all-cause trauma (not only work injuries) [19]. While it is difficult to interpret the difference, the prior study focused on self-rated pain and disability scores as the outcomes where herein we see the largest difference when self-rated work status is the outcome. We also acknowledge the large number of analyses for which the significance threshold (p) was not adjusted, raising the possibility that this is an alpha error. Regardless of the mechanism, the findings of the current study have opened the possibility that males may require a different prognostic tool than females, which if true is the first time such a finding has been shown to our knowledge and creates space to question the results of prior work on prognostic tools. Similarly, the EQ-5D-5L results may signal something about the items in that scale or its scoring algorithm that lend to its robustness across demographic strata, from which other prognostic tools may benefit.

There are important limitations to consider. This work was conducted within a single Canadian province, with rehabilitation provided by clinics within a single large network, and under the funding models and guided by protocols of a single rehabilitation funder. While these helped strengthen some aspects of internal validity, they also provide questions on generalizability that would benefit from work outside of these jurisdictions. As mentioned earlier, the sociodemographic variables were dichotomized into groups that are arguably overly broad and may not hold intuitive meaning. Our selection of thresholds on variables other than sex was largely driven by the data, using median splits as a meaningful distinction that also permitted interpretable AUC analyses. Arguments could easily be made for choosing different thresholds, for creating more categories (though sample sizes would get quite small in some cells) or for leaving age and InDI-M score in their raw ordinal or continuous forms. Some of these decisions were *a priori* while others were *a posteriori* when the lower-than-expected rates of recovery were identified. We acknowledge that dichotomizing variables can obscure some (non)-linear effects, and expect that with larger samples there will be opportunities to explore finer gradations in differential effects across levels of constructs like age, income, or education. We also note that sample size likely played a role in some of the findings being just shy of Z-test significance (e.g., NPRS AUC when self-reported work status was the outcome). A larger sample would likely have reduced the width of the confidence intervals and may have led to more significant findings. We also highlight the exploratory nature of the analyses of mean baseline values of the three predictors, for which we did not correct the alpha error rate based on multiple analyses. This could have led to some means being significantly different by chance, though the only finding that this could have affected was the TIDS x InDI-M categories, for which a familywise Bonferroni corrected p value would have made that difference not statistically significant. The NPRS x Age comparison would remain significant, and there were no other significant findings, rendering this a mild concern. The almost fully self-reported nature of the data introduces both strengths and limitations, a strength in that the results are

patient-centred (reported directly and independently by the patient) without risk of observer bias, though limited as these were completed without oversight meaning we cannot ensure whether adequate attention was paid or that the questions were properly interpreted. This is a common limitation of survey-based research, and the ability to also mine the administrative database, which was created through opinions of clinicians, funders, and administrators, was a useful and different source of information for these analyses.

## Conclusions

The purpose of this hypothesis-generating study was to explore the ways that different sociodemographic identities influence either the scores expected on three common clinical self-report tools, or the ability of those tools to predict outcomes after an 8-week standardized rehabilitation program. The results indicate that when using NPRS as a potential risk/prognosis screening tool after acute work injuries consideration should be given to using different cut scores for older and younger respondents, and not at all depending on the outcome being predicted. Scores on the TIDS, a tool specifically intended for risk/prognosis in MSK trauma, were only different across levels of prior experiences of intersectional discrimination which supports Cumulative Stress Theory but is unlikely to be easily implemented in practice. Recovery rates differed considerably based on the definition used, from 21.7% to 54.7%, highlighting the importance of careful selection of recovery indicators in prognostic research. Finally, both the TIDS and EQ5D-5L showed consistent and largely similar ability to discriminate between those who would and would not eventually report recovery, though findings provided some evidence of differential functioning of the TIDS by sociodemographic categories that may suggest a different prognostic/risk screening tool be considered for some groups, such as males. We are unaware of prior work on any prognostic screening tool for MSK pain that has explored these differential effects. Several new lines of inquiry have been proposed as a result of these findings.

## Supporting information

**S1 Table. Proportion of respondents selecting each ethnoracial identity category.** Respondents were free to select as many categories as applies to them.
(DOCX)

## Acknowledgments

We thank the staff and clients at CBI Health Inc. in Ontario for their contributions to the success of this study.

## Author contributions

**Conceptualization:** Tom Carter, David M. Walton.

**Data curation:** David M. Walton.

**Formal analysis:** Mohammad Bayattork, Saghar Soltan Abadi, David M. Walton.

**Funding acquisition:** David M. Walton.

**Investigation:** David M. Walton.

**Methodology:** Mohammad Bayattork, Tom Carter, David M. Walton.

**Project administration:** Mohammad Bayattork, David M. Walton.

**Resources:** Lynn Cooper, David M. Walton.

**Software:** David M. Walton.

**Supervision:** Mohammad Bayattork, Tom Carter, Lynn Cooper, David M. Walton.

**Validation:** Tom Carter, Lynn Cooper, David M. Walton.

**Writing – original draft:** Mohammad Bayattork, Saghar Soltan Abadi, David M. Walton.

**Writing – review & editing:** Mohammad Bayattork, Saghar Soltan Abadi, Tom Carter, Lynn Cooper, David M. Walton.

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
