## [Decision Letter · Decision Letter 0]

1 Dec 2025

Dear Dr.  Walton,

Thank you for submitting your manuscript to PLOS ONE. After careful consideration, we feel that it has merit but does not fully meet PLOS ONE’s publication criteria as it currently stands. Therefore, we invite you to submit a revised version of the manuscript that addresses the points raised during the review process.

We look forward to receiving your revised manuscript.

Kind regards,

Nadinne Alexandra Roman, Ph.D.

Academic Editor

PLOS ONE

Journal Requirements:

“Ontario Workplace Safety and Insurance Board (WSIB) Research Grant awarded to the senior author (Walton), held at Western University in London, Ontario. Funds were dedicated to reimbursement of study participants, offsetting costs of data extraction, aggregation, and sharing through the CBI Health Group Inc. administrative team, and acknowledgement of the time on project for patient partner L. Cooper.”

3. Please note that your Data Availability Statement is currently missing or the DOI/accession number of each dataset OR a direct link to access each database. If your manuscript is accepted for publication, you will be asked to provide these details on a very short timeline. We therefore suggest that you provide this information now, though we will not hold up the peer review process if you are unable.

Reviewer's Responses to Questions

**Comments to the Author**

1. Is the manuscript technically sound, and do the data support the conclusions?

Reviewer #1: Yes

Reviewer #2: Yes

Reviewer #3: Partly

2. Has the statistical analysis been performed appropriately and rigorously?

Reviewer #1: Yes

Reviewer #2: Yes

Reviewer #3: Yes

3. Have the authors made all data underlying the findings in their manuscript fully available?

Reviewer #1: Yes

Reviewer #2: Yes

Reviewer #3: Yes

4. Is the manuscript presented in an intelligible fashion and written in standard English?

Reviewer #1: Yes

Reviewer #2: Yes

Reviewer #3: Yes

Reviewer #1: This manuscript presents a well-designed and methodologically rigorous study investigating the prognostic utility of three commonly used clinical self-report tools (NPRS, TIDS, and EQ-5D-5L) in adults with acute musculoskeletal (MSK) injuries, with a focus on potential differences in functioning across sociodemographic subgroups. Overall, the study is technically sound, and the data support the conclusions drawn.

Strengths:

Novelty and Significance: The study addresses an important gap in prognostic research by examining how social and demographic factors may influence both baseline scores and predictive accuracy of clinical tools, which has implications for personalized rehabilitation. The inclusion of both self-reported and administrative recovery outcomes adds depth and robustness to the findings.

Methodology: The study uses validated instruments, clearly defined outcomes, and appropriate statistical analyses, including AUC analyses for discriminative accuracy. Data cleaning and handling of missing responses are described transparently.

Clarity of Reporting: The manuscript is well-organized, with clear presentation of results in tables and figures. Discussion thoughtfully interprets the findings in light of previous literature, acknowledges limitations, and proposes directions for future research.

Ethical and Research Integrity: There are no apparent concerns regarding dual publication, research ethics, or data integrity. Consent procedures and data handling are appropriately described.

Areas for Consideration / Minor Suggestions:

While sociodemographic variables were dichotomized for analysis, further discussion of potential limitations of broad categorizations (e.g., “non-white”) is included in the manuscript. Future work could explore finer-grained categories with larger samples.

The sample size for some subgroup analyses is relatively small, which may limit the power to detect differences. The authors acknowledge this and interpret findings cautiously.

The manuscript could benefit from a brief note on the clinical implications of the findings, particularly regarding how clinicians might use TIDS or EQ-5D-5L scores in practice to inform individualized prognostic assessments.

Conclusion:

Overall, this manuscript represents a well-conducted and timely contribution to the literature on prognostic tools in acute MSK injury and their interaction with sociodemographic factors. The findings are novel, methodologically sound, and the conclusions are appropriately supported by the data. I consider this manuscript suitable for publication.

Reviewer #2: This is an exceptionally well-designed and clearly presented study that addresses an important and timely issue in musculoskeletal injury care. The authors chose appropriate, rigorous methods and presented them transparently, with thoughtful attention to prognostic validity and recovery definition. The integration of multiple recovery indicators, the exploration of subgroup effects, and the use of ROC/AUC approaches all contribute meaningful insight to the literature. The manuscript is logically structured, easy to follow, and clinically relevant, and the findings have clear implications for improving risk stratification and patient management in work-related MSK injuries.

I have two minor suggestions.

One is to consider whether any of the dense data presented in table format might be presented graphically in a figure. If this is not practical given the complexity of the data, the manuscript will work as is.

The second is to consider expanding the discussion point about the finding that high TIDS scores may signal underlying cumulative stress and discrimination that can impede recovery. One potential actionable takeaway might be that clinicians caring for those with a high TIDS score include strategies to identify and address cumulative psychosocial load, not just treat the MSK injury. This may be an interesting avenue for follow-on research.

Overall, this is high-quality work, and I congratulate the authors on a strong contribution to the field. I support acceptance of the manuscript.

Reviewer #3: This manuscript addresses an important and underexplored question at the intersection of prognostic research, work-related MSK injury, and social inequities. The dataset and open data sharing are valuable strengths. With revisions to strengthen the statistical rigor and more cautious, clearly exploratory framing of subgroup findings—especially regarding dichotomization, multiple comparisons, power, and generalizability—the study could make a useful contribution to the literature on personalized and equitable prognosis in injured workers.

Major Comments:

1. Overall technical soundness and conclusions:

The prospective cohort design and choice of measures are appropriate, and the main descriptive findings (e.g., older workers reporting higher pain, higher discrimination associated with greater distress, and TIDS/EQ-5D-5L generally outperforming NPRS) are broadly supported by the data. However, several analytic choices reduce statistical rigor and the precision of inferences. In my view, the conclusions—especially about differential prognostic performance by subgroup and implications for cut-scores—should be more clearly framed as exploratory and hypothesis-generating rather than confirmatory.

2. Statistical analysis: dichotomization and collapsed categories:

A central concern is the extensive dichotomization and collapsing of variables:

Age, income, and InDI-M are split at the median into “high/low” groups.

Education is collapsed to “university degree vs. no university degree”.

Race/ethnicity is collapsed to “White-identifying vs. non-White-identifying”.

While this may have been motivated by power considerations for subgroup ROC analyses, dichotomization is known to reduce power, obscure non-linear relationships, and increase the risk of misleading findings. The very broad “non-White” category is particularly problematic in an equity-focused study, as it likely masks heterogeneity among minoritized groups.

Suggestions:

Retain age, income, and InDI-M as continuous predictors in at least some models (e.g., logistic regression for recovery outcomes), and clearly label subgroup analyses as secondary/exploratory.

Provide more detailed racial/ethnic category frequencies in a table or supplement, even if inferential analyses must use a collapsed variable.

Strengthen the Discussion’s acknowledgement that these broad categories—especially “non-White”—limit interpretability and do not provide definitive evidence for any specific racialized group.

3. Multiple comparisons and exploratory nature of subgroup AUCs:

The study includes a large number of hypothesis tests: baseline group comparisons, multiple recovery indicators, and extensive subgroup ROC/AUC analyses (Tables 4–6). There is no formal adjustment for multiple testing, and the implications of this for Type I error are only briefly mentioned. This is important for results that are highlighted in the Discussion, such as higher TIDS scores in those with higher discrimination and sex differences in TIDS performance.

Suggestions:

State explicitly in the Methods that analyses are exploratory and not adjusted for multiple comparisons.

In the Results/Discussion, emphasize that significant subgroup effects (e.g., TIDS × sex) should be interpreted cautiously in light of multiplicity and limited power.

If feasible, consider a simple sensitivity analysis (e.g., FDR) for key sets of tests and comment on whether main conclusions are robust.

4. Power and precision of subgroup analyses:

Although you exclude subgroup ROC analyses with very small cells, many of the reported subgroup AUCs still have wide confidence intervals and non-significant between-group Z-tests. At times the Discussion appears to give qualitative emphasis to differences that are statistically non-significant and imprecise.

Please make clearer in the text that these subgroup comparisons are underpowered and that the study was not designed to definitively test interaction effects (e.g., sex × tool, age × tool). Where specific differences are highlighted (such as better TIDS performance in females), present them explicitly as hypotheses for future, larger studies rather than firm conclusions.

5. Recovery definitions and discrepancies:

Using three distinct recovery indicators (self-rated global recovery, self-reported work status, and administrative discharge status) is a major strength and reveals substantial differences in “recovery” rates depending on how recovery is defined. This deserves more explicit discussion.

I encourage you to further reflect on:

How administrative “full recovery” (driven by clinician/funder judgement) may be influenced by non-clinical factors.

What it means for prognostic validation when patient self-report and administrative outcomes diverge (i.e., which outcome should a “good” prognostic tool prioritise?).

Clarifying in the Methods how ambiguous administrative cases were coded would also be helpful.

6. Generalizability and context:

The study is conducted in a single jurisdiction, provider network, and funding scheme (WSIB PoC in Ontario). This context likely limits generalizability to other compensation systems, uninsured or informally employed workers, and different health systems. Although some of this is acknowledged, I recommend slightly strengthening the Discussion to more clearly state these external validity limitations.

7. Conflict of interest and tool ownership:

The competing interests statement is transparent about the developer of the TIDS being an author and about the tool’s open license. Given that TIDS is one of the focal tools and generally performs well, a brief reminder of this relationship in the Methods or Discussion, coupled with a note that analyses followed pre-specified methods and that raw data are publicly available, would further reinforce transparency.

Minor Comments:

1. Study period:

The Results report recruitment as “From October 2023 to May 2025”, which appears to be a typographical or dating error. Please verify and correct the study dates.

2. “Cross-sectional and longitudinal” wording:

It may help readers if you explicitly state early in the Methods that the design involves a baseline assessment with an 8-week follow-up, clarifying that the longitudinal component is relatively short.

3. Language and readability:

The manuscript is clearly written and in standard English. A light edit at revision could shorten some long sentences in the Discussion and check for minor typographical and numerical inconsistencies.

4. Tables 4–6:

These tables are dense. Consider improving readability by clearly marking which AUCs are significantly >0.5 and which between-group differences are significant, and by visually highlighting only those subgroup effects you intend to emphasize in the text.

.

Reviewer #1: **Yes:** Mohammad FayazMohammad FayazMohammad FayazMohammad Fayaz

Reviewer #2: No

Reviewer #3: No

---

## [Author Response · Author response to Decision Letter 1]

18 Dec 2025

We have uploaded a full point-by-point response to each reviewer comment as part of the resubmission documents.

---

## [Editor Report · Decision Letter 1]

24 Mar 2026

Social determinants of pain, distress, and quality of life in injured workers: A cross-sectional and longitudinal analysis of patient-reported outcomes

PONE-D-25-51049R1

Dear Dr. Walton,

We’re pleased to inform you that your manuscript has been judged scientifically suitable for publication and will be formally accepted for publication once it meets all outstanding technical requirements.

Kind regards,

Nadinne Alexandra Roman, Ph.D.

Academic Editor

PLOS One
---

## [Editor Report · Acceptance letter]

PONE-D-25-51049R1

PLOS One

Dear Dr. Walton,

I'm pleased to inform you that your manuscript has been deemed suitable for publication in PLOS One. Congratulations! Your manuscript is now being handed over to our production team.

Kind regards,

on behalf of

Dr. Nadinne Alexandra Roman

Academic Editor

PLOS One